# Correlates of disease severity in bluetongue as a model of acute arbovirus infection

**Vanessa Herder** [1], **Marco Caporale**[2], **Oscar A. MacLean**[1], **Davide Pintus**[3], **Xinyi Huang**[1], **Kyriaki Nomikou**[1†], **Natasha Palmalux**[1], **Jenna Nichols**[1], **Rosario Scivoli**[3], **Chris Boutell**[1], **Aislynn Taggart**[1], **Jay Allan**[1], **Haris Malik**[1], **Georgios Ilia**[1], **Quan Gu**[1], **Gaetano Federico Ronchi**[2], **Wilhelm Furnon**[1], **Stephan Zientara**[4], **Emmanuel Bréard**[4], **Daniela Antonucci**[2], **Sara Capista**[2], **Daniele Giansante**[2], **Antonio Cocco**[2], **Maria Teresa Mercante**[2], **Mauro Di Ventura**[2], **Ana Da Silva Filipe**[1], **Giantonella Puggioni**[3], **Noemi Sevilla**[5], **Meredith E. Stewart**[1], **Ciriaco Ligios**[3], **Massimo Palmarini**[1] *

1 MRC-University of Glasgow Centre for Virus Research, Glasgow, United Kingdom, 2 Istituto Zooprofilattico Sperimentale dell' Abruzzo e Molise "G. Caporale", Teramo, Italy, 3 Istituto Zooprofilattico Sperimentale della Sardegna, Sassari, Italy, 4 Laboratory for Animal Health, INRAE, Ecole Nationale Vétérinaire d'Alfort, ANSES, Maisons-Alfort, France, 5 Centro de Investigación en Sanidad Animal. Instituto Nacional de Investigación y Tecnología Agraria y Alimentaria. Consejo Superior de Investigaciones Científicas (CISA-INIA-CSIC). Valdeolmos, Madrid, Spain

† Deceased.
* massimo.palmarini@glasgow.ac.uk

**Data Availability Statement:** All code used to analyse the data is available at https://github.com/omaclean/sheep_ML (see the readme for specific scripts). Raw data underpinning the figures of this

## Abstract

Most viral diseases display a variable clinical outcome due to differences in virus strain virulence and/or individual host susceptibility to infection. Understanding the biological mechanisms differentiating a viral infection displaying severe clinical manifestations from its milder forms can provide the intellectual framework toward therapies and early prognostic markers. This is especially true in arbovirus infections, where most clinical cases are present as mild febrile illness. Here, we used a naturally occurring vector-borne viral disease of ruminants, bluetongue, as an experimental system to uncover the fundamental mechanisms of virus-host interactions resulting in distinct clinical outcomes. As with most viral diseases, clinical symptoms in bluetongue can vary dramatically. We reproduced experimentally distinct clinical forms of bluetongue infection in sheep using three bluetongue virus (BTV) strains (BTV-$1_{IT2006}$, BTV-$1_{IT2013}$ and BTV-$8_{FRA2017}$). Infected animals displayed clinical signs varying from clinically unapparent, to mild and severe disease. We collected and integrated clinical, haematological, virological, and histopathological data resulting in the analyses of 332 individual parameters from each infected and uninfected control animal. We subsequently used machine learning to select the key viral and host processes associated with disease pathogenesis. We identified and experimentally validated five different fundamental processes affecting the severity of bluetongue: (i) virus load and replication in target organs, (ii) modulation of the host type-I IFN response, (iii) pro-inflammatory responses, (iv) vascular damage, and (v) immunosuppression. Overall, we showed that an agnostic machine learning approach can be used to prioritise the different pathogenetic mechanisms affecting the disease outcome of an arbovirus infection.

study are presented either as Supplementary Tables or are deposited in the Elighten repository (http://dx.doi.org/10.5525/gla.researchdata.1622). The raw FASTQ files associated with this project have been submitted to the European Nucleotide Archive (ENA; project accession number PRJEB72808).

**Funding:** This study was funded by the Wellcome Trust (206369/Z/17/Z to MP; https://wellcome.org/) and in part by the EU Horizon 2020 (H2020 PALE-Blu grant project No: 727393-2 to MP; https://research-and-innovation.ec.europa.eu/funding/funding-opportunities/funding-programmes-and-open-calls/horizon-2020_en), the Italian Ministry of Health (RC IZS SA 02/16 to GP, CL, RS, DP and RC IZS SA 04/18 16 to GP, CL, RS, DP; https://www.salute.gov.it/portale/home.html), a Research Fellowship by the Deutsche Forschungsgemeinschaft (DFG; Project number 406109949 to VH; https://www.dfg.de/en) and the Medical Research Council (MRC, MC_UU_00034/5 to MP; https://www.ukri.org/councils/mrc/). VH received a salary from the DFG funding. The funders did not play any role in the study design, data collection and analysis, decision to publish, or preparation of the manuscript.

**Competing interests:** The authors have declared that no competing interests exist.

## Author summary

In this study we comprehensively investigated the pathogenetic mechanisms underlying the clinical outcomes of bluetongue, a viral disease of ruminants used as an experimental model for vector-borne infections. Arboviruses are the cause of major global health and economic burden. Each arbovirus infection induces its own distinctive clinical features. However, many of these vector-borne diseases are typically characterised by general symptoms such as mild febrile flu-like illness and rash, with only a minor proportion of cases exhibiting severe clinical manifestations. It is therefore critical to understand those biological processes distinguishing arbovirus infections resulting in mild or severe clinical diseases. Here, we used *in vivo* experiments in sheep, artificial intelligence and *in vitro* experiments to identify the key processes affecting the severity of bluetongue. We established that the clinical outcome of bluetongue in the infected animals is influenced by (i) levels of virus replication in target organs, (ii) modulation of the host innate immunity and (iii) pro-inflammatory responses, (iv) damage to the blood vessels, and (v) immuno-suppression. This study provides an intellectual framework on how to prioritise experimentally the variety of biological processes determining the clinical outcome to a virus infection.

## Introduction

Viral diseases are characterised by a wide spectrum of clinical symptoms that can vary substantially in their disease severity and pathogenesis. Understanding which complex virus-host interactions determine the clinical outcome of infections is the cornerstone of viral pathogenesis. Certain factors defining virus and host responses influencing disease severity are well understood [1]. For example, virus strains or variants with inherent higher or lower virulence have been described for many viruses including human or avian influenza viruses, SARS-CoV-2, dengue virus, foot and mouth disease virus, and many others [2–5]. Individual host susceptibility to virus infections is also affected by a variety of factors including age, genetic variability, pre-existing immunity, co-infections or other co-morbidities [6,7]. For example, genetic variations in genes associated with the interferon response can be responsible for more severe manifestations of acute respiratory infections such as influenza and COVID-19 [8–12]. Polymorphism in chemokine receptors instead can slow progression of HIV-1 infection [13,14].

Observational studies or genome wide association studies with very large clinical cohorts are needed to correlate disease severity with distinct aspects of host responses or individual genetic differences and biomarkers [15]. These studies are feasible for diseases with a high disease burden such as COVID-19, influenza, and HIV/AIDS. However, in general, it is difficult to systematically investigate *in vivo* the fundamental aspects of the complex virus-host interactions underlying different clinical outcomes of infection. This is particularly true for many arbovirus infections, where the most common clinical outcome is febrile illness that rarely progresses to severe disease, and it is therefore mostly undiagnosed. In chikungunya virus infection for example, only a small subset of patients progress to a chronic infection and develop arthritis, which is associated with increased levels of serum cytokines and impaired immune cell functions [7]. Progressing to the severe haemorrhagic form of dengue fever virus infections is caused by the presence of anti-dengue virus antibodies from previous infections leading to a more severe disease phenotype caused by antibody-dependent enhancement [16].

Animal models are extremely useful to understand virus pathogenesis, as the various stages of disease progression can be investigated longitudinally from the very early sub-clinical incubation period to late times post-infection. Rodent models, and the mouse in particular, have been extensively used to study viral pathogenesis. For example, gene knock-out mice have been instrumental to understand many aspects of the innate and adaptive immune response to virus infections [17,18]. However, in most cases, disease pathogenesis in experimental animal models do not fully reflect the true co-evolutionary interactions between a virus and its natural host.

In order to define the correlates of disease severity of acute arbovirus infections, we studied the pathogenesis of bluetongue, a vector-borne disease of domestic and wild ruminants. Bluetongue is caused by bluetongue virus (BTV), a dsRNA Orbivirus within the family of the *Sedoreoviridae* [19–22] transmitted by the biting midge (*Culicoides* spp.). Bluetongue is endemic in most continents, and there are at least 35 BTV serotypes circulating worldwide [23,24]. BTV-8 for example, caused a major epizootic in Europe between 2006 and 2010, while BTV-3 has re-emerged very recently in the Netherlands causing major problems to animal health [25]. The re-emergence of bluetongue in Northern Europe is a stark reminder of the geographical expansion of vector-borne diseases in the last two decades [25,26].

Bluetongue is an excellent model to study the pathogenesis of acute arbovirus infections due to its wide diversity in clinical outcomes, offering a unique opportunity to dissect the intricate process governing disease severity and progression. First, the disease in sheep is characterised by a wide spectrum of clinical outcomes, ranging from a mild febrile illness to a lethal disease characterised by oedema, haemorrhages, and respiratory symptoms [27–31]. Secondly, bluetongue can be reproduced experimentally in sheep reflecting natural disease outcomes observed in the field within the same host [32,33]. Here, we aimed to investigate the complex virus-host interactions leading to different clinical trajectories in bluetongue, as a model for acute arbovirus disease. We developed an experimental framework using three different strains of BTV resulting in different clinical outcomes in sheep, ranging from clinically unapparent disease to mild and severe disease. We then used supervised machine learning [34] to evaluate more than 330 individual parameters related to virus replication and associated host responses to infection. Using this approach, we define and validate the pathways and biomarkers associated with different clinical trajectories of disease in mammalian hosts in response to arbovirus infection.

## Results

### Experimental reproduction of divergent clinical outcomes of bluetongue infection

To investigate the pathogenesis of acute arbovirus infection, we experimentally infected sheep with either one of the following BTV strains: BTV-1$_{IT2006}$, BTV-1$_{IT2013}$, BTV-8$_{FR2017}$ [35,36]. We chose these three viruses as they were associated with field outbreaks of bluetongue with distinct clinical outcomes: BTV-1$_{IT2006}$ induced severe clinical disease, BTV-8$_{FR2017}$ caused a clinically unapparent or mild disease, while sheep infected with BTV-1$_{IT2013}$ showed an intermediate phenotype [35,36].

We attempted as much as possible to maintain "natural" conditions in our experimental model. Therefore, we isolated these viruses directly from blood of infected animals and passaged them no more than three times in a *Culicoides* cell line to minimise mutations brought about by cell culture adaptation. We infected male and female Sarda sheep by intradermal inoculation to "mimic" an insect bite. We carried out animal experiments in two distinct locations (S1 Table) using either male (location 1) or female (location 2) sheep. Henceforth, each

animal experimental group will be described by "G1" or "G2" to indicate the location of the experiment, followed by the virus used for the experimental infection. For example, the group of sheep infected with BTV-1$_{IT2013}$ in location 1 are defined as sheep "G1-BTV-1-2013". Infected animals were observed for 7 days, corresponding to the peak of clinical signs (with the exception of G1-BTV-1-2013 which were kept for 21 days). We used 7 animals for each group and included a mock-infected control group (matched for age, breed, and sex) at each location. Body temperature and clinical signs were recorded daily and compiled in clinical scores (as described in Methods and previously [37]). Although clinical scores (with the exception of fever) are predominantly based on visual examination, which can be difficult to clearly discern between intermediate phenotypes, the disease in rams in G1 typically displayed higher clinical scores, higher fever and more severe clinical manifestations than in G2. Animals G1-BTV-1-2006 displayed the most severe clinical signs; G2-BTV-8 showed the mildest outcome, while sheep infected with BTV-1$_{IT2013}$ showed an intermediate phenotype. Overall, we were able to reproduce experimentally the clinical outcome that was also described in the field.

Infection in sheep with BTV-1$_{IT2006}$ was characterised by severe subcutaneous oedema (Fig 1a) and acute respiratory symptoms with heavy breathing that presented with ulceration and erosion of the nostrils (Fig 1b) relative to control animals (Fig 1c), as well as severe systemic signs of infection including high fever. Female sheep infected with BTV-1$_{IT2006}$ showed a less severe disease outcome as well lower rectal temperature compared to rams infected with BTV-1$_{IT2006}$ (Fig 1d–1f). To confirm this apparent sex bias, we experimentally infected three additional male sheep with BTV-1$_{IT2006}$ at location G2 (S1 Fig). In general, rectal temperatures in infected sheep peaked at 6 to 7 days post infection (dpi; Fig 1d and 1e). Male sheep infected with BTV-1$_{IT2006}$ in location G2 displayed a more severe clinical disease than female animals

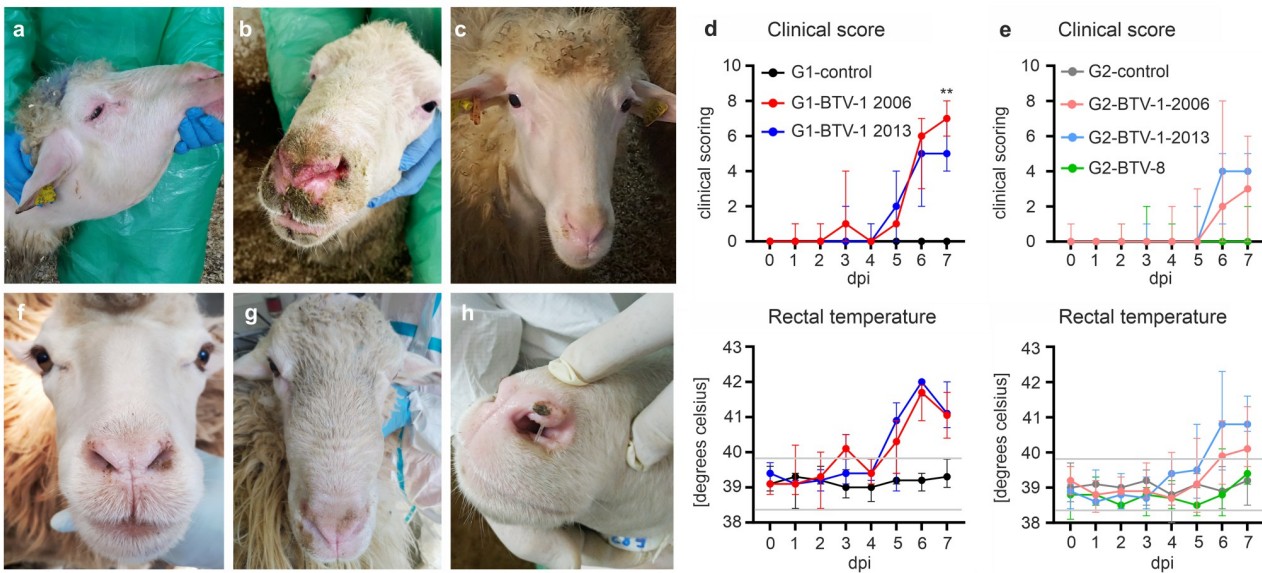

**Fig 1. Distinct clinical phenotypes in experimentally induced bluetongue.** (a-c) Images of sheep experimentally infected with different strains of BTV in location G1. (a-b) Rams within the G1-BTV-1-2006 group displaying severe subcutaneous oedema in the neck (a), or ulcerations and crusts of the nostrils and surrounding skin (b). (c) Healthy mock-infected G1-control ram. (d-e) Clinical score and rectal temperature of all experimentally infected sheep and control used in this study. (f-h) Images of sheep experimentally infected with different strains of BTV in location G2. (f) Mock-infected female G2-control sheep. (g) Female sheep (G2-BTV-1-2006) with a moderate subcutaneous oedema of the head and nose bridge. (h) Mild, focal ulceration on the nostril of a female sheep (G2-BTV-1-2013. 2-way ANOVA, ** = p<0.01. Data are shown as the median, with minimum and maximum observed values. Grey lines indicate the upper and lower reference value for the physiological rectal temperature in sheep.

in the same location (S1 Fig). These data suggest that sex, not location, underlies the differences in disease severity observed in groups G1 and G2.

Male and female sheep infected with BTV-1$_{IT2013}$ (G1-BTV-1-2013 and G2-BTV-1-2013) showed respiratory symptoms including ulceration of the nasal mucosa, nasal discharge, coughing, and a moderate subcutaneous oedema that was typically associated with increased rectal temperatures (Fig 1d and 1e). Male sheep in G1 showed more severe clinical signs of infection when infected with BTV-1$_{IT2006}$ compared to animals infected with BTV-1$_{IT2013}$ (p = 0.0072) at 7 dpi (Fig 1d). However, the increase in rectal temperature of the same animals did not differ significantly at 7 dpi (Fig 1e). Female sheep infected with BTV-8$_{FRA2017}$ (G2-BTV-8) displayed clinically unapparent or very mild clinical signs, with a minimal increase in rectal temperatures (Fig 1d and 1e).

## Machine learning approaches to identify correlates of disease severity

From each experimentally infected and control group we collected blood (at day 0, 1, 3, and 7 days post infection, dpi). In addition, several tissues were collected during post-mortem examination at 7 dpi (tongue, lung, skin at the site of the experimental infection, skin at a distal site of the experimental infection, and lymph nodes draining the inoculation site) from all groups with the exception of G1-BTV-1-2013. Samples were used for the following analyses: whole blood transcriptome, serum cytokines, blood biochemistry, viremia, and serology. In addition, using quantitative immunohistochemistry of post-mortem tissues, we assessed the distribution of BTV antigens (NS2) in the tissues listed above. We also evaluated the number of follicles, and T and B cells (including Foxp3 regulatory T cells) in the lymph nodes draining the site of virus inoculation. Analysis and comparison of whole blood transcriptome between animal groups was facilitated using integrated gene sets differentiated into blood transcriptional modules (BTMs) which have been previously identified in both humans and sheep [38,39]. Hence, we collected a total of 332 parameters and established a supervised machine learning (ML) approach aimed to identify the correlates of viral pathogenesis and clinical trajectories of disease outcome (S2 Table).

We simultaneously analysed the data by grouping datasets in three alternative ways, according to number and type of variables: (i) 6 "states of infection", (ii) 4 "states of infection" or (iii) "clinical states". The "6 states of infection" analysis considered two variables, the virus strain and location/sex: G2-BTV-1-2006 (female); G1-BTV-1-2006 (male), G2-BTV-1-2013 (female), G2-BTV-8 (female) as well as control animals G1-control (male) and G2-control (female). The 7 male sheep G1-BTV-1-2013 were excluded as no post-mortem tissue from these animals could be collected at 7 dpi. The 6 "states of infection" represents the most stringent way of grouping the infected animals as they are differentiated on the bases of both the inoculated virus, sex and location of the experiment. The 4 "states of infection" analysis considered a single variable, the virus strain, irrespective of their sex or location of the experiment: G1/G2-control (male and female), G1/G2-BTV1-2006 (male and female), G2-BTV1-2013 (female only) and G2-BTV8 (female only; Fig 2b). In the "clinical states" analysis, animals were grouped on the bases of their clinical score (irrespective of the virus used for infection) that was arbitrarily differentiated into: no clinical signs (score 0), mild disease (scores 1 to 2), moderate disease (scores 3 to 5) and severe disease (scores 6 to 8) irrespective of the virus, location or sex of the experimentally infected sheep.

We then utilised the random forest machine learning approach to analyse the data arranged in each of the groupings described above. We used recursive feature elimination to find the most predictive core subset of parameters distinguishing each group. We selected arbitrarily the number of parameters based on where the slope of the prediction accuracy curve

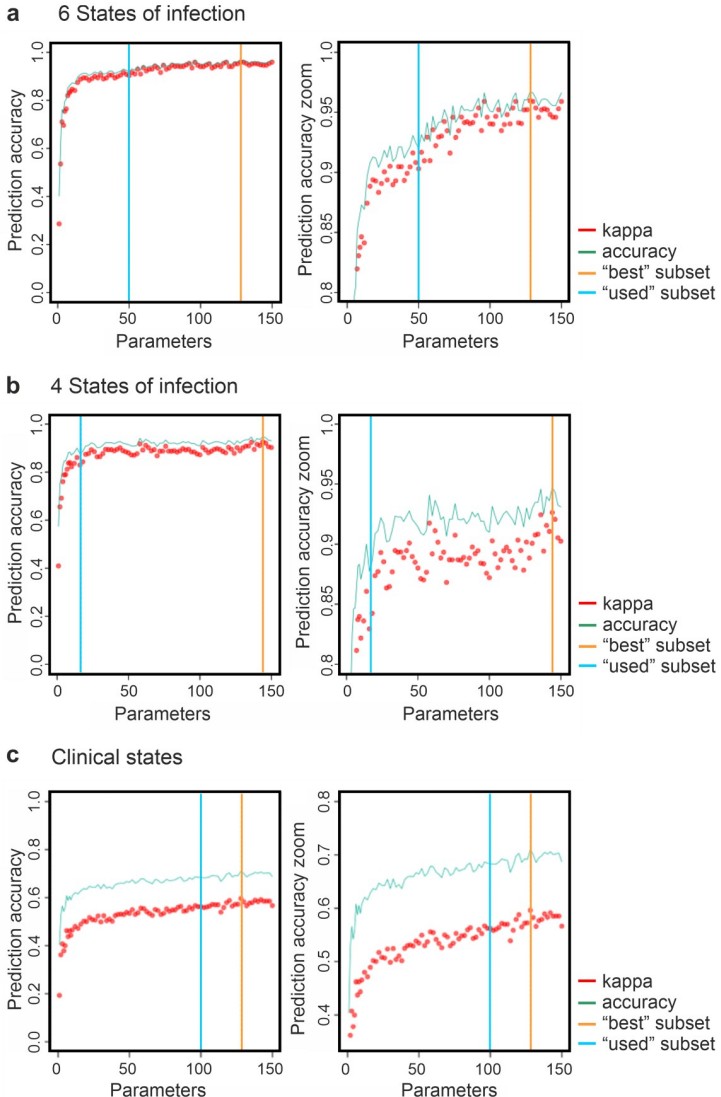

**Fig 2. Selection of core subsets of predictive correlates of disease severity.** Using 332 parameters obtained from each animal in this study, we used a supervised machine learning approach to identify the most predictive core subset of parameters distinguishing each group. The datasets were grouped in three distinct ways: (a) 6 states of infection (groups distinguished on the bases of virus and location used in this study), (b) 4 states of infection (groups distinguished on the bases of the virus used only), and (c) clinical states (groups distinguished on the bases of their clinical scores only). 50 parameters show a prediction accuracy of more than 90% for the 6 states of infection. 17 parameters show a prediction accuracy of more than 90% for the 4 states of infection, while 100 parameters show a prediction accuracy of more than 65% for the clinical states of infection. Note full y-axis on the left side, while the right side is zoomed in for clarity. The accuracy for the clinical scores is likely inflated by the imbalanced group sizes, thus the greater discrepancy between accuracy and kappa. Accuracy = proportion of samples correctly assigned; kappa = Cohen's Kappa: adjusted prediction accuracy to reflect the expected performance of random guesses & account for imbalanced classes; best subset = highest accuracy of any individual run; used subset = selected number of parameters for further analyses.

plateaued, indicative of redundancy of additional parameters (Fig 2a–2c; Tables 1 to 3). To accurately predict the six states of infection we required a minimum of 50 out of the 332 predictive parameters, which we prioritised based on their "gini-importance" value (Fig 2a; S3 Table). This value is obtained from the sum of the number of sample splits across all decision

**Table 1. Prediction values for machine learning analysis with "6 states of infection" using Random Forest.**

| Groups* | G2-control | G2-BTV-1-2013 | G2-BTV-1-2006 | G2-BTV-8 | G1-control | G1-BTV-1-2006 |
|---|---|---|---|---|---|---|
| G2-control | 999 | 1 | 0 | 0 | 0 | 0 |
| G2-BTV-1-2013 | 39 | 824 | 14 | 123 | 0 | 0 |
| G2-BTV-1-2006 | 0 | 0 | 1000 | 0 | 0 | 0 |
| G2-BTV-8 | 0 | 0 | 0 | 1000 | 0 | 0 |
| G1-control | 18 | 0 | 0 | 0 | 982 | 0 |
| G1-BTV-1-2006 | 0 | 0 | 0 | 2 | 0 | 998 |

* 1000 repeated random forest cross validations; real states are in the rows and the predictions are the columns.

**Table 2. Prediction values for machine learning analysis with "4 states of infection" using Random Forest.**

| Groups* | G1-/G2-control | G1-/G2-BTV-1-2013 | G1-/G2-BTV-1 2006 | G1-/G2-BTV-8 |
|---|---|---|---|---|
| G1-/G2-control | 987 | 0 | 13 | 0 |
| G1-/G2-BTV-1-2013 | 0 | 874 | 16 | 110 |
| G1-/G2-BTV-1 2006 | 80 | 21 | 899 | 0 |
| G1-/G2-BTV-8 | 26 | 29 | 0 | 945 |

* 1000 repeated random forest cross validations; real states are in the rows and the predictions are the columns.

"trees" that include the feature, proportional to the number of samples it splits [40]. Hence, this gini-importance value represents the relative importance of each parameter for classification in the random forest.

We used 1000 rounds of cross validation, each using randomly selected 5 animals from each class to train the model and 2 unseen animals to test the performance of the model. With this approach, four of six infectious states (G2-BTV-1-2006, G2-BTV-8, G1-BTV-1-2006 and G2-control) were identified with > 98% accuracy (Table 1). Infection with G2-BTV-1-2013 was predicted correctly in 82% of the cases, while the G1-control group in 98.2% of cases. The strong sex/location signal in the data are highlighted by the complete separation of the two control groups (in the ML predictions; Table 1).

Given the strong separation of uninfected male and female controls in the two locations, we sought to find parameters which predicted the infecting strain across both male and female sheep to determine generalisable trends. We therefore applied our random forest approach to the dataset arranged in the four states of infection (Fig 2b). We found that the four groupings were accurately predicted using only 17 of the 332 parameters (Fig 2b; Tables 2 and S4). The highest accuracy was reached with the control sheep (987/1000) containing in this instance

**Table 3. Prediction values for machine learning analysis with "clinical states" of infection using Random Forest.**

| Groups* | Control | Scores 0–2 | Scores 3–5 | Scores 6–8 |
|---|---|---|---|---|
| Control | 851 | 146 | 1 | 2 |
| Scores 0–2 | 178 | 479 | 339 | 4 |
| Scores 3–5 | 39 | 207 | 694 | 60 |
| Scores 6–8 | 129 | 0 | 46 | 825 |

* 1000 repeated random forest cross validations; real states are in the rows and the predictions are the columns.

both male and female animals from both study sites. Animals G2-BTV-8 were predicted correctly in 94.5% of the times. The most severe clinical phenotype G1-G2-BTV1-2006 was detected accurately in more than 89.9% of the time (899/1000). The lowest accuracy was evident for sheep infected with G2-BTV-1 2013 (874/1000) with overlap into other virus-infected groups (Table 2).

We then applied the same approach to the "clinical states" of infection, where animals were divided simply on the bases of the symptomatology displayed (no symptoms, mild, moderate, and severe as recorded by the clinical scores) irrespective of the virus used for the experimental infections and/or sex. The most accurate prediction was reached with 100 predictive parameters (Fig 2c). We obtained relatively high prediction values within the control groups (851/1000) and animals with severe disease (825/1000; Tables 3 and S5). As expected, we found lower prediction accuracy on the groups with a relatively mild or intermediate disease phenotype, as they are understandably difficult to separate on the bases of observational scores only. The low clinical score (0 to 2) was predicted correctly in 479/1000 tests while the moderate clinical scores in 694/1000, with the latter showing substantial overlaps into the groups with mild and high scores, as well as non-infected control animals.

We also run the Boruta algorithm [41] for 5000 iterations to confirm the random forest approach: 17 out of 17 parameters from the four state list are in the 60 parameter confirmed list from Boruta (S6 Table); 50/50 parameters from the six state parameter list are in the 93 parameter confirmed list from Boruta (only 1 parameter was tentative for this model) (S7 Table); less parameters (39/100) from the clinical score list are in the confirmed list of 64 parameters from Boruta (S8 Table). These low values reflect also the low predictive values of the parameters identified by random forest as the "clinical states" are defined by the clinical scores of the infected animals, which in turn are based mostly on subjective values of the observers.

In addition, to further confirm our ML results were robust and not due to overfitting on background noise, we trained random forests on shuffled data tables (confusion matrices), where each sample's data was randomly re-assigned, removing the link between infection state, and collected data. The prediction accuracy on this shuffled dataset was lost, demonstrating the robust nature of our methodology, picking out genuine signals from the data rather than simply over-fitting (S9, S10 and S11 Tables). ROC (receiving operating characteristics) curves, plotting the type I and type II error trade-offs in our imperfect predictive signals, are shown in S3 Fig.

## Key processes distinguishing the clinical trajectory of bluetongue

We next focused our study on those parameters with the highest gini-importance values that were common predictors for at least two of the three groupings described above. This unbiased approach identified in total 35 parameters (Fig 3 and S12 Table), and among these, 8 (Fig 3a, arrows) were shared by all 3 approaches. Overall, from these parameters, we identified five different fundamental processes in virus-host interactions affecting the clinical trajectory of bluetongue: (i) virus load, (ii) the host type-I interferon (IFN) response; (iii) pro-inflammatory responses, (iv) vascular damage, and (v) immunosuppression (Fig 3b and 3c).

## Virus load and replication in target organs

Blood viremia, assessed as the amount of viral RNA in the blood, was the parameter with the highest impact on the severity of the disease identified by our machine learning approach (Fig 3a). We detected viral RNA in the blood of some infected animals from 1 dpi. By 3 dpi most infected animals, except G2-BTV-8 tested positive for viral RNA. All animals, with the

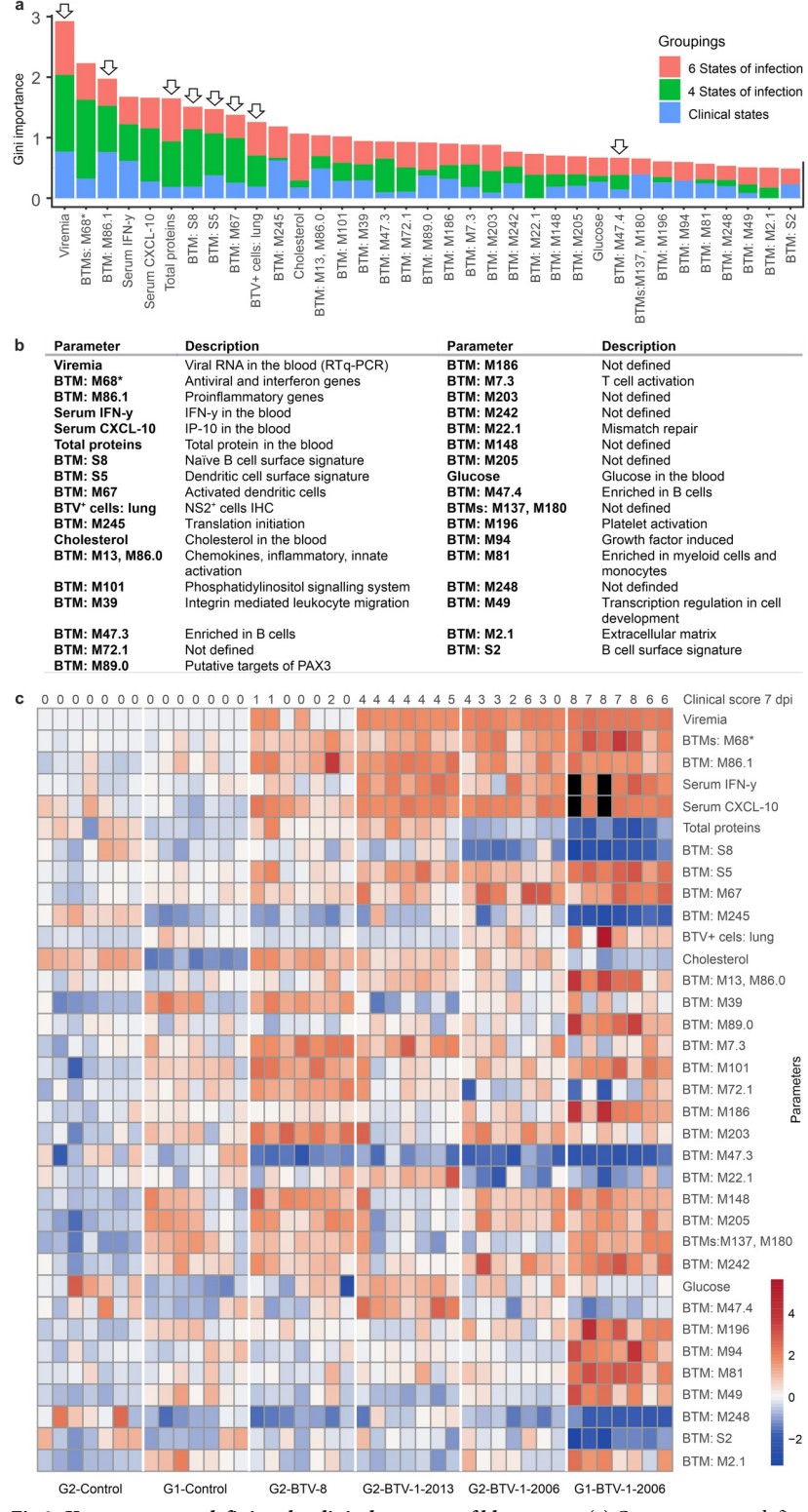

**Fig 3. Key parameters defining the clinical outcome of bluetongue.** (a) Core parameters defined as those with the highest gini-importance number that are common to at least two of the three groupings analysed in Fig 2. Arrows indicate the 8 parameters identified in all three groupings. (b) Brief description of the core parameters shown in a. Note that M68 (identified by an asterisk) is merged with similar BTMs containing overlapping sets of genes. (c) Heatmap showing the relative quantification of the 35 parameters divided by group and individual animals used in this

study. Black boxes indicate when the values were not available. Top numerical row indicates the clinical scores of individual animals.

exception of 4 BTV-8 infected sheep and the control groups, became viraemic at 7 dpi (Figs 3c and 4a). Importantly, we detected the highest values of viral RNA at 7 dpi in the blood in those G1-BTV1-2006 animals with the most severe clinical symptoms, highlighting a positive correlation between viremia and severity of clinical signs (Fig 4a).

The relative amount of virus antigen in the tissues of infected animals at 7 dpi was also one of the 35 key parameters distinguishing clinical outcome (Fig 3b). We assessed BTV replication in the tissues of infected animals by quantifying NS2-positive signal by immunohistochemistry. We applied software-assisted unbiased image analysis on stained tissue from tongue, lung, skin (both at the sites of virus inoculation and in a distant site), and in the lymph nodes draining the site of inoculation. In general, tongue and lung from BTV-infected animals showed the highest relative values of viral protein expression (Fig 4b–4h). Impairment of these organs is consistent with the dominant clinical symptoms of bluetongue in sheep including lesions to the tongue and respiratory distress. We detected BTV antigen and RNA in the endothelial cells of the tongue by both immunohistochemistry and RNA *in situ* hybridisation, respectively (Fig 4c–4h). The skin of animals with bluetongue also showed erythema. In the lung, bronchial and alveolar epithelial cells were positive for viral antigen. At 7 dpi in the BTV-infected animals, G1-BTV-1-2016 demonstrated the highest proportion virus-infected cells than sheep in groups G2-BTV-1-2013 and G2-BTV-8 (Fig 4b). We found virus in the endothelial cells of small blood vessels of both the skin corresponding to the sites of virus inoculation, and in distant areas. These data suggest that the virus infects the endothelial cells of the deep dermis after the viraemic phase. However, confocal microscope analysis of additional skin samples at the site of virus inoculation, collected at earlier time points (2 dpi) from animals infected with BTV-$1_{IT2006}$ (included in a separate pilot experiment), showed BTV to mainly infect the endothelial cells of the lymphatic vessels (Fig 4i).

Overall, these data show that the severity of disease is correlated to BTV infection of endothelial cells in peripheral organs including skin, lungs, and tongue which are reached by the virus during the viraemic phase of disease progression.

## Timing of the type-I IFN response

The second parameter with the highest gini-importance value, among the 35 identified above (Fig 3a), were BTMs which included genes involved in the type-I interferon (IFN) response (BTM M68 and others; S12 Table). These BTMs include transcription factors such as IRF-7 and interferon stimulated genes like CXCL10, ISG15, IFIT-1, IFIT-2, RSAD2, OAS1 among others. Other BTMs highlighting innate immune activation include those associated with dendritic cell activation (BTM M67), and general chemokine, inflammatory and innate activation (BTMs M13 and M86.0). Importantly, analysis of blood transcriptome using standard pathway analysis methods also revealed "cytokine signalling in immune system" (GO:0019221), "innate immune responses" (GO:0002226) and "interferon signalling" (GO:0060337) as the strongest upregulated pathways discriminating the different clinical outcomes of bluetongue infection (S2 Fig; S13 and S14 Tables).

Overall, the data described above indicate that the type-I IFN response is a key correlate of disease outcome. Hence, we further analysed the transcriptional profile of interferon stimulated genes (ISGs), which collectively orchestrate the host antiviral state, in the blood of infected and uninfected animals from the early phase of the infection, to the peak of clinical manifestations (1 to 7 dpi, respectively) to capture the temporal modulation of the IFN

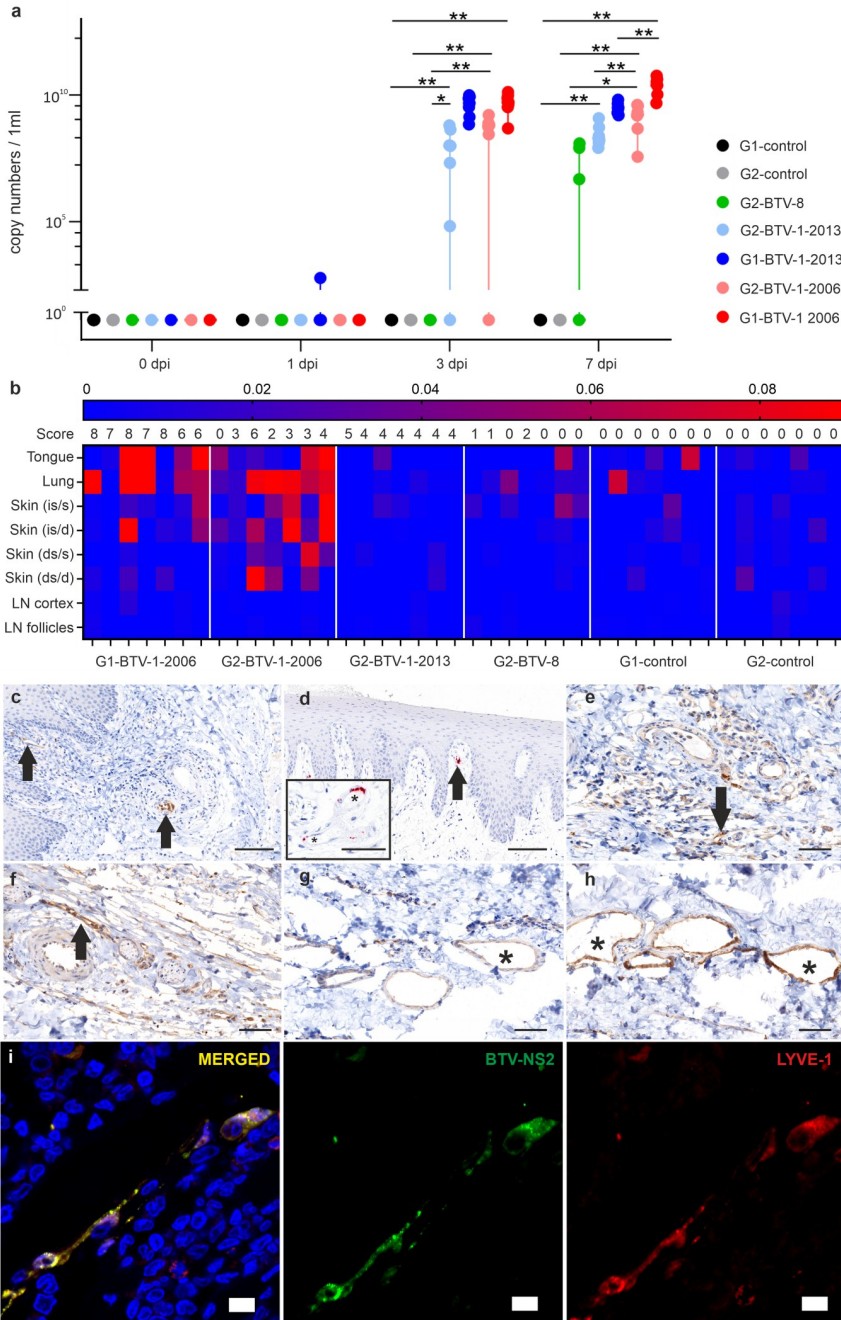

**Fig 4. Viral replication in blood and tissues of infected animals.** (a) Scatter dot plot showing levels of BTV RNA obtained by qRT-PCR in blood samples. Data indicate medians with minimum and maximum values. Statistical significance between groups was measured by unpaired t-test (* is p = <0.05, ** is p = <0.01). (b) Heatmap displaying the relative amount of BTV NS2 protein in tissues collected from infected animals at 7 days post-infection. Values were obtained by relative quantification of positive signal from immunohistochemistry of whole tissue section slides and downstream software-assisted image analysis as described in Methods. Data are normalised to average obtained in control animals. (c) Immunohistochemistry of tissue section from the tongue of a sheep within the G1-BTV-1-2006 group showing viral NS2 (brown signal, arrows) in endothelial cells of arteries and veins (bar = 100 μm). (d) *In situ* hybridisation of tissue section from the tongue of a sheep within the G1-BTV-1-2006 group revealing the presence of viral RNA in endothelial cells of an artery (red signal, arrow); Asterisk in the insert highlights more endothelial cells with a positive signal for viral RNA. Bar insert = 50 μm; bar main panel = 100 μm. (e-h) Immunohistochemistry in tissue sections derived from skin of infected animals at the inoculation sites showing viral NS2 (brown signal, arrow) in endothelial cells. Images representative of skin samples from animals in group G1-BTV-1-2006 (e), G2-BTV-1-2006

(f), G2-BTV-1-2013 (g); G2-BTV-8 (h); Bars = 60 μm. (i) Confocal microscope images from skin (inoculation site) tissue sections of a sheep infected with BTV-1$_{IT2006}$ collected at 2 dpi. Virus NS2 (green) is detected in lymphatic endothelial cells infected cells as highlighted with antibody towards Lyve-1 (red). Cell nuclei are shown in blue (bars = 10 μm).

response. Animals infected with BTV-8 (which presented a very mild to asymptomatic infection) showed the strongest early upregulation of ISGs (1 dpi; Fig 5a). Indeed, it is notable that we could detect upregulated BTMs associated with innate immune response (and pro-inflammatory response–see below) in all animals infected with BTV-8$_{FR2017}$, although viremia was only detected in 3 of the 7 infected sheep.

Conversely, sheep within the G1-BTV-1-2006 group, with the severe clinical phenotype, showed instead no upregulated ISGs at 1 dpi (Fig 5a), while animals infected with BTV-1$_{IT2013}$ with an intermediate phenotype showed mild upregulation of ISGs. At 3 and 7 dpi instead, sheep within the groups with most severe disease (G1-BTV-1-2006) showed the strongest ISG induction, while it was minimal in G2-BTV-8 animals. These data suggest that replication of the less virulent virus BTV-8$_{FR2017}$ was controlled by the host type-I IFN response at the early stages post-infection, while the more virulent BTV1$_{IT2006}$ was able to block the early IFN response *in vivo*.

To corroborate these data, we infected ovine endothelial cells with the viruses used in this study and analysed the ISG response at 6 and 12 hpi *in vitro*. Similarly, to what we observed *in vivo*, at early times post-infection, BTV-1$_{IT2006}$ induced a relatively reduced ISG response compared to BTV-8 (Fig 5b). These data suggest that the most virulent virus, BTV-1$_{IT2006}$, is better equipped to modulate the IFN response compared to the BTV-8$_{FR2017}$.

## Pro-inflammatory response leading to vascular damage

The next three parameters with the highest gini-importance values were all related to the host pro-inflammatory response. BTM 86.1 was the parameter with the third highest gini-importance value, and included many proinflammatory mediators such as IFNγ, TNF, CCL4, CCL20 and NFKB1. This BTM was more strongly upregulated in sheep with more severe disease compared to those in the G2-BTV-8 group and the mock-infected controls. Importantly, these data extracted from RNAseq, were confirmed by Luminex assays highlighting a significant upregulation of cytokines including IFN-γ and CXCL10/IP-10 (fourth and fifth highest gini-importance values) in the sera of sheep with more severe and moderately serious clinical signs, compared to groups with milder disease (Fig 6a). Other BTMs among the 35 key predictive parameters of disease severity included M13 and M86.1 that overlap with M86.1 above and genes associated with the innate immune response and proinflammatory mediators such as IL1β. Overall, data obtained from both RNAseq and Luminex assays demonstrate disease severity is to be tightly associated with the induction of pro-inflammatory cytokines.

The presence of pro-inflammatory mediators in the blood can contribute to vascular damage. Correspondingly, the sixth predictive parameter of BTV disease severity was the amount of total blood proteins detected in infected animals. Animals with severe bluetongue showed a profound loss of total proteins in the blood, which is a consequence of vascular damage (Fig 6b). Furthermore, genes associated with platelet activation (BTM M196) were also strongly upregulated in severe cases of bluetongue (G1-BTV-1-2006). Furthermore, RNAseq pathway analysis also confirmed genes associated with vascular disease to be more significantly upregulated in severely ill sheep (S2c Fig).

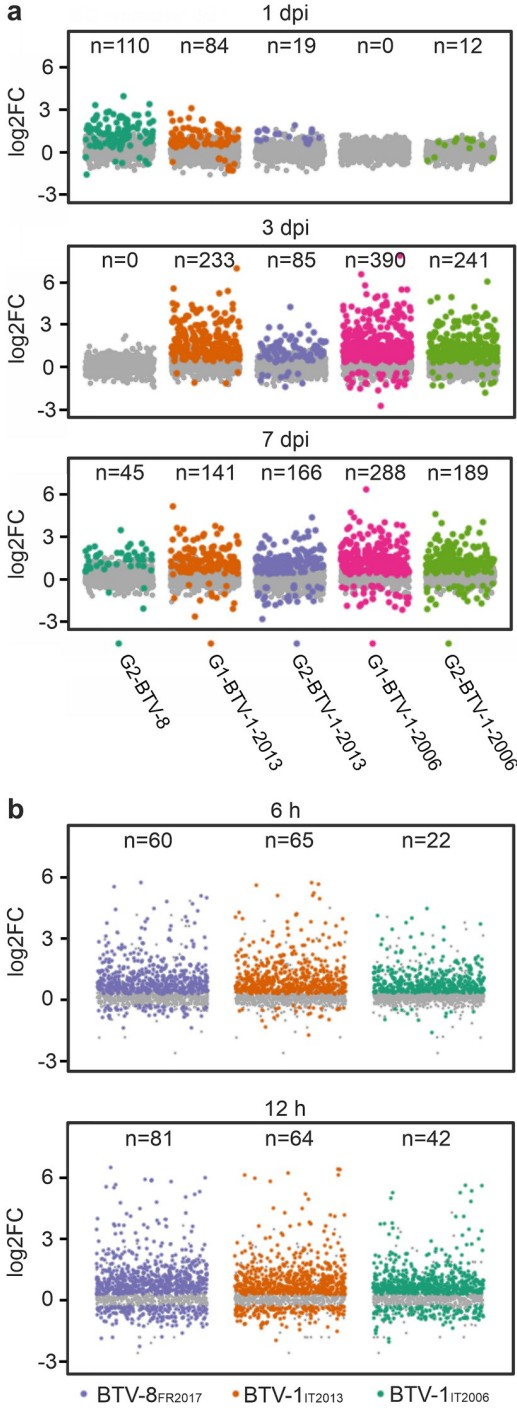

**Fig 5. Early modulation of ISGs upregulation correlates with disease severity.** (a) Scatter dot plots illustrating systemic ISGs upregulation in infected animals (normalised to mock-infected controls). Data were obtained by RNAseq from the blood at 1, 3 and 7 days post infection (dpi). Numbers on top of each graph illustrate the number of significantly upregulated ISGs (FDR < 0.05), showed as bright coloured dots compared to grey background). Note that animals in location G1 infected with BTV-1$_{IT2006}$ show no ISG upregulation at early timepoints (1 dpi), while show the

largest number of differentially regulated ISGs at later time points. (b) Scatter plots showing upregulation of ISGs in endothelial cells infected *in vitro* with either BTV-8_{FR2017}, BTV-1_{IT2013}, or BTV-1_{IT2006} compared to uninfected mock controls. Numbers on top of each graph illustrate the number of significantly upregulated ISGs (FDR < 0.05), showed as bright coloured dots compared to grey background).

### Immunosuppression

The next parameter with the highest gini-importance values was BTM S8, which is associated with naïve B cell surface signatures. Importantly these were reduced in sheep with more severe disease. Notably, two other BTMs associated B-cell enrichment (BTM M47.3 and BTM M47.4) were also identified in the top 35 parameters suggesting a trend in B cell reduction in animals with severe disease. These data are in line with previously published reports showing that BTV causes lymphopenia [42] and a transitory immunosuppression due to inhibition of B-cell division in germinal centres due to infection of follicular dendritic cells [32,43]. Indeed, we detected a reduced number of follicles in the lymph nodes draining the sites of virus inoculation in animals with severe disease (Fig 6d). We also confirmed lymphopenia in G1-BTV-1-2006 animals compared to control animals, although for logistic reasons we could not perform this particular experiment in animals in the G2 group (Fig 6c). Importantly, we identify BTM 7.3, associated with genes related to T-cell activation, to be significantly upregulated in animals with no (G2-BTV-8) or moderate disease relative to severe disease (G1-BTV-1-2006). In addition, we further analysed regulatory T cells (Treg), as they are key drivers in the modulation of antiviral immunity. We detected a significant reduction of Foxp3$^+$ Tregs in the lymph nodes of G1-BTV-1-2006 animals compared to BTV-8_{FR2017} and BTV-1_{IT2013}-infected ones at 7 dpi (Fig 6e). The reduction of Tregs in severely diseased animals suggests an important role of this T cell subgroup in the outcome of BTV pathogenesis, which are known to control virus-induced tissue damage and dampen overwhelming immune responses [44,45]. Overall, immunosuppression was also confirmed by RNAseq pathway analysis, which demonstrated significantly higher levels of differential gene expression in G1-BTV-1-2006 (S2d Fig).

### Discussion

In this study we comprehensively investigated the pathogenic mechanisms underlying the clinical outcomes of arbovirus infection. Arboviruses are the cause of major global health and economic burden [46]. While each arbovirus has its own distinctive clinical features, many of these vector-borne diseases in humans (and animals) are typically characterised by general symptoms such as mild febrile flu-like illness and rash, with only a minor proportion of cases exhibiting severe clinical manifestations [47,48]. It is therefore critical to understand the underlying biological processes responsible for the varied pathogenic outcomes to arbovirus infection and their relative importance to disease outcome in a systematic manner.

In this study, we used bluetongue, a major vector-borne disease of ruminants, as an experimental model in its natural host species. We were able to experimentally recapitulate the varied clinical outcomes of BTV infection in sheep, ranging from unapparent or mild febrile symptoms (BTV-8_{FR2017}) to more severe disease (BTV-1_{IT2006}), including respiratory distress, subcutaneous haemorrhage, oedema, and tongue lesions. We collected data encompassing 332 biological parameters related to both virus and host from infected or control animals, and used machine learning to prioritise the key correlates of disease severity.

Machine learning is being increasingly used in the infectious disease field to find applicable biomarkers of disease trajectories [49–51]. In most cases, however, this approach relies on disease systems trained on data obtained from large patients cohorts or previously characterised patients [34]. Here, we used machine learning in an unbiased approach to identify the key

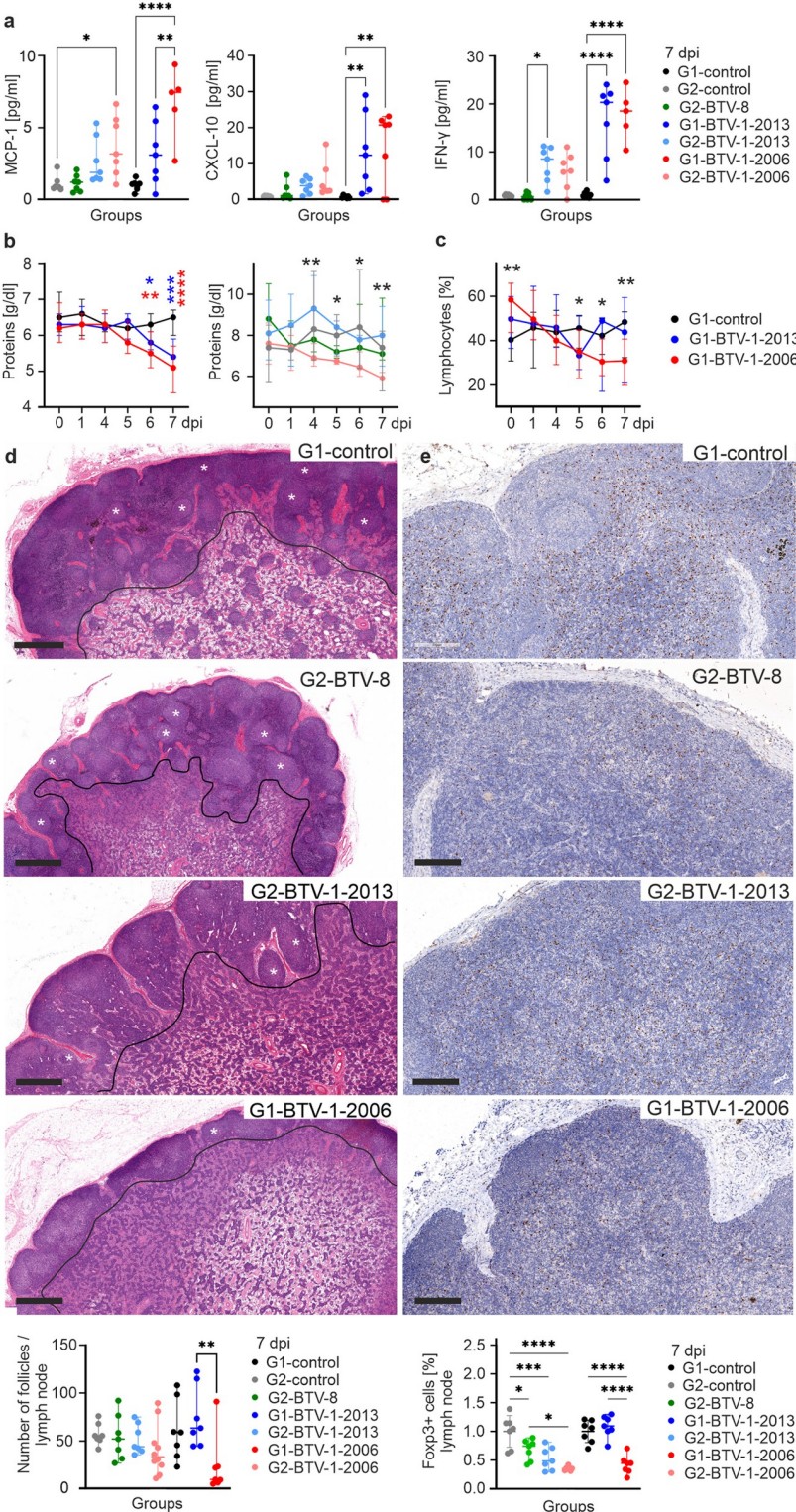

**Fig 6. Extent of pro-inflammatory response, protein loss and lymphopenia correlate with disease severity.** (a) Levels of pro-inflammatory markers (MCP-1, CXCL-10/IP-10, IFN-ɣ) in the serum of infected and mock-infected animals at 7 days post-infection (dpi). Data for each animals are normalised to levels at day 0. (b) Level of total proteins in the sera of infected and mock-infected controls during the course of the experiment. Red and blue asterisks indicate statistical differences compared to mock. For the statistical differences in location G2 the asterisks indicates multiple

comparisons: 4 dpi = G2-BTV-1-2013 vs. G2-BTV-1-2006 p = 0.0024, G2-BTV-1-2013 vs. G2-BTV-8 p = 0.0158, G2-BTV-1-2006 vs. G2-BTV-8 p = 0.0143; 5 dpi = G2-BTV-1-2013 vs. G2-BTV-1-2006 p = 0.0286, G2-BTV-1-2013 vs. G2-BTV-8 p = 0.0233; 6 dpi = G2-BTV-1-2013 vs. G2-BTV-1-2006 p = 0.0298; 7 dpi = G2-control v. G2-BTV-1-2006 p = 0.0234; G2-BTV-1-2013 vs. G2-BTV-1-2006 p = 0.0019, G2-BTV-1-2006 vs. G2-BTV-8 p = 0.0205. (c) Lymphopenia shown as reduced lymphocyte counts in the blood of sheep with severe disease (G1-BTV-1-2006) compared to mock-infected controls. Black asterisks indicate differences between G1-BTV-1-2006 and mock. (d) Top panels, photomicrographs of lymph node sections stained with haematoxylin and eosin and quantification. Some follicles are highlighted with asterisks. Black line indicates the border between the lymph node cortex and medulla. Bars = 1 mm. Bottom panel, number of follicles in the cross-section of a lymph node draining the site of virus inoculation. Significant difference is shown between G1-BTV-1-2006 and G1-BTV-1-2013. (e) Top panels, immunohistochemistry of lymph node sections stained with an antibody for the nuclear marker Foxp3. Bars = 200µm. Bottom panel, measurement of Foxp3-positive cells by software-assisted image analysis in lymph node sections. All graphs: Data are shown as the median, with minimum and maximum observed values. 2-way ANOVA, * = p<0.05; ** = p<0.01; *** = p<0.001; **** = p<0.0001.

drivers of the pathogenesis of an arbovirus infection. Our approach identified the (i) levels of virus replication in the infected host, (ii) timing of the host innate immune response, (iii) levels of pro-inflammatory mediators, (iv) vascular damage, (v) and immunosuppression to be critical in contributing functions of disease progression and severity (Fig 7).

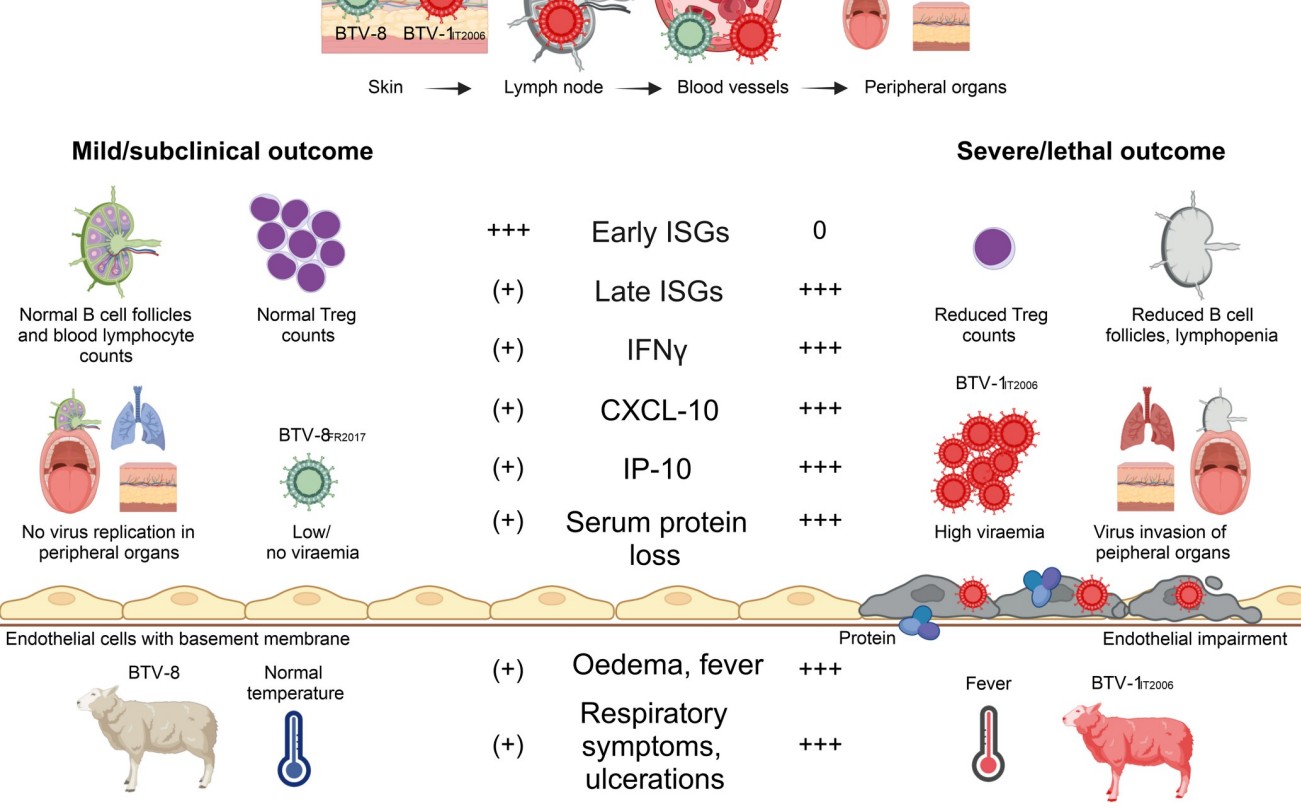

**Fig 7. Schematic representation of the key pathogenetic mechanisms of bluetongue pathogenesis.** The virus replicates in the sites of virus inoculation and then in the regional lymph nodes, before entering the blood compartment and reaching the peripheral organs. Animals with severe disease are characterised by a late IFN response, high proinflammatory mediators, reduced blood proteins, high viraemia and viral replication in peripheral organs. Created in Biorender.

Viral load in the blood was the highest parameter that directly correlated to disease severity. Levels of viremia were related to the detection of virus in endothelial cells in the lung, tongue, and skin of infected sheep, which directly recapitulate the clinical signature of bluetongue: respiratory distress, lesions in the tongue epithelium, and sub-cutaneous haemorrhage.

The second prioritised parameter by our experimental approach was related to the type-I IFN response, one of the key host antiviral innate immune responses. Resolving viral infection in mammals is associated with an effective "antiviral state", which is orchestrated by the production of interferons inducing the activation of hundreds of interferon-stimulated genes (ISGs) [52]. At 7 dpi (peak clinical symptoms), a stronger systemic IFN response correlated with more severe disease. However, at early time points (1 dpi) there was an inverse correlation. We detected a stronger systemic type-I IFN response in sheep infected with the attenuated BTV-8$_{FR2017}$ (G2-BTV-8 group) compared to those with a more severe disease. This trend was confirmed *in vitro*, where cells infected with BTV-1$_{IT2006}$ showed reduced ISG expression at 6 hpi compared to cells infected with BTV-8$_{FR2017}$. We conclude that more virulent BTV strains appear to be efficient at modulating the type-I IFN response, allowing the virus to reach higher viral loads and further dissemination to target organs. These data are consistent with our recent findings in BTV infected primary bovine cells [53]. Cows are known reservoirs of BTV infection but rarely show clinical disease. Comparison of a virulent strain of BTV in ovine and bovine cells identified the type-I IFN response to be initiated earlier in bovine cells leading to a greater restriction in virus replication [53]. BTV modulates the interferon response through the expression of non-structural proteins with immunomodulatory functions, such as NS3 and NS4, and by inducing host cell translational shutdown [54–58]. It is most likely that BTV strain- and/or species-specific variance in the expression and/or biological activity of these viral immune evasion agonists account for the differences observed in type-I IFN response.

In addition, we found a high pro-inflammatory immune signature to correlate with disease severity. We identify both the transcript levels of inflammatory genes and pro-inflammatory cytokine mediators (IFN-γ and CXCL-10/IP-10) to strongly correlate with disease progression. CXCL10/IP-10 expression has been shown to be induced by IFN-γ in a range of cell types, including leukocytes, neutrophils, eosinophils, endothelial cells, and monocytes, and to correlated with disease severity [59]. Virus replication in endothelial cells, but more importantly a systemic pro-inflammatory immune response can contribute to increased capillary permeability, vascular leakage and therefore loss of blood protein in interstitial spaces. Indeed, we found that the reduction of total blood proteins in infected animals to be directly related to disease severity. Vascular damage and hypoproteinaemia with oncotic pressure, lead to oedema, which is one of the key symptoms that we detected in sheep with more severe clinical disease (Fig 1). Vascular leakage is also a pathological feature severe of dengue virus infection [60].

Like many acute infections, BTV is initially controlled by the type-I IFN response prior to induction of the adaptive immunity. We did not systematically characterise the host adaptive immune response in this study, but we could detect (i) a reduction of B-cell signatures in the blood transcriptome, and (ii) a reduction in the number of B-cell follicles in the regional lymph nodes draining the sites of virus inoculation in sheep affected by more severe disease. These data support previously published studies identifying peripheral B-cell reduction in BTV-infected animals [61], and demonstrating BTV disruption of follicular dendritic cells the in regional lymph nodes close to the site of inoculation, leading to a block in B-cell maturation and immunosuppression [62]. In addition, animals within the G1-BTV-1-2006 group showed a marked leukopenia (compared to G1-control and G1-BTV-1-2013), which is a common feature in BTV-infected animals [63]. Overpowering immune responses and virus elimination are controlled by regulatory T cells (Tregs) [64]. Tregs during virus infection limit the

efficiency of anti-viral immunity and tissue damage due to excessive inflammation. Here, we identify a severe disease phenotype of BTV infection in sheep to substantially reduced Treg-numbers in the draining lymph nodes. These data suggest that a prolonged impairment of Tregs during acute BTV-infection may have a substantial negative impact upon the outcome of the disease, as previously described for other infections [65,66–68].

The data we obtained also suggest that rams display a worse clinical outcome to BTV infection than ewes, however further experimental studies will be required to conclusively address this point. Usually, in domestic flocks there are several dozen ewes for each ram, hence it is difficult to extrapolate solid data with regards to sex-bias from the naturally occurring outbreaks of bluetongue.

In summary, our systematic experimental approach, and the use of machine learning, has identified the key parameters of an arboviral disease progression. We showed how bluetongue disease severity is directly linked to immunopathology, which in turn is governed by the balance between virus replication, and the host innate and adaptive immune responses. Early activation of the type-I IFN response strongly correlates to reduced viral load, reduced pro-inflammatory responses, and a milder clinical outcome of infection. On the other hand, late IFN responses are instead associated with high viral load, high pro-inflammatory responses, vascular damage, and immunosuppression leading to more severe disease outcomes (Fig 7). Our study provides an overall framework to understand and compare the pathogenesis and disease progression of arbovirus infections.

## Materials and methods

### Ethical statement

Animal experiments carried out in this study were approved by the ethical committees of the Istituto Zooprofilattico della Sardegna and Istituto Zooprofilattico Sperimentale dell'Abruzzo e Molise, and further authorised by the Italian Ministry of Health in accordance with EU laws 26/2014 (permission numbers 797/2018-PR and 688/2018-PR).

### Cells and viruses

The three viruses used in this study, BTV-1$_{IT2006}$, BTV-1$_{IT2013}$ and BTV-8$_{FR2017}$ originated from blood of infected sheep [35,36] and were isolated and passaged up to three times in KC cells [69]. These cells are derived from *Culicoides sonorensis*, and were grown at 28˚C using Schneider's insect media (Sigma-Aldrich) with 10% foetal calf serum (FCS; ThermoFisher). Supernatants were harvested 5 days post infection (dpi) and virus titres determined by standard plaque assays in CPT-Tert cells as previously described [70,71]. As a control, supernatant of non-infected KC cells diluted with PBS was used. CPT-Tert cells [72] derive from sheep choroid plexus cells immortalized with simian virus 40 (SV40) T antigen and human telomerase reverse transcriptase, and were propagated in Iscove's modified Dulbecco's medium (IMDM) supplemented with 10% FBS. Immortalised primary ovine endothelial cells were used as described previously [53]. For the RNAseq experiments described below, immortalised ovine endothelial cells were infected in 12-well plates at MOI~10 with either BTV-1$_{IT2006}$, BTV-1$_{IT2013}$ or BTV-8$_{FR2017}$ by spinoculation at 500 x g at 4˚C for 1 h. The inoculum was aspirated, cells washed twice with warm media supplemented with 10% FCS and incubation at 37˚C in a humidified 5% $CO_2$ environment for either 6 or 12 hpi. After incubation, monolayers were washed with warm PBS to remove any cellular debris and immediately lysed in 750 μL of Trizol. Total RNA was extracted from each sample as previously published [53].

## Animals and study design

Sarda sheep were kept in mosquito-proof, air-controlled facilities with *ad libitum* access to tap water to acclimatise before being enrolled in the experiments. Animals were fed pellets and hay daily. 59 adult sheep (Sarda breed) were infected with BTV-1$_{IT2006}$, BTV-1$_{IT2013}$, BTV-8$_{FR2017}$ or mock-infected, respectively (S1 Table). Experiments were carried out in two distinct locations (G1 and G2). Rams were used in location G1, while ewes in location G2. In the location G1, in total 21 male Sarde sheep were inoculated (mock, BTV-1$_{IT2006}$, BTV-1$_{IT2013}$, n = 7, respectively). Sheep with mock and BTV-1$_{IT2013}$ were sacrificed at 21dpi, while BTV-1$_{IT2006}$ animals were culled between 7 and 9 dpi due to welfare reasons. In the location G2, in total 28 female sheep (mock, BTV-1$_{IT2006}$, BTV-1$_{IT2013}$, BTV-8$_{FR2017}$; n = 7 per group) were kept for 7 dpi. Two additional experiments were carried out in location G2: 3 male sheep were infected with BTV-1$_{IT2006}$ in location G2 and culled at 7 dpi. In addition, 4 female sheep were infected with BTV-1IT2006, or mock-infected, and killed at two days post-infection (S1 Table).

Each animal was inoculated intradermally with multi-site injection in 4 distinct areas (500μl per inoculation area) in proximity of the right axillary, left axillary, right inguinal and left inguinal lymph nodes. In total, we used $2x10^5$ pfu of virus per animal. Control animals received the same treatment with the same volume of diluted, mock-infected Kc cell culture supernatant. Animals were killed at 7 dpi, with the exception of 4 animals infected with BTV-1$_{IT2006}$ in location G1 which were killed at 8 or 9 dpi. As highlighted above, three additional rams were used in location G2 to control for sex differences in the severity of BTV-1$_{IT2006}$ infection. In addition, to control for the early sites of viral replication in the skin, 4 ewes and mock-infected controls were killed at 2 dpi. Animals used in the experiment were screened for the presence of antibodies towards BTV [73] and for viral RNA in the blood by qRT-PCR (see below). Only seronegative and PCR negative animals were used in the study. Animals were dewormed with netobimin 4ml/10kg body weight. Blood for haematology was collected only from animals in location G1. Sera were instead collected for serology, blood chemistry and cytokines analysis (from both animals in G1 and G2). For RNAseq, 2.5ml blood from animals in G1 and G2 was collected in PAXgene Blood RNA Tubes (IVD, Qiagen, 762165). Rectal temperature and clinical score were assessed by 2 veterinarians daily according to the previously published scoring index [37]. Tissues samples were collected at post-mortem and fixed in 4% buffered formalin (FFPE), or PAXgene fixative (PFPE, PAXgene Tissue FIX Container; 50 ml; Qiagen, 765312), for 16 hours and subsequently embedded in paraffin wax. All paraffin wax embedded tissue blocks were stored at 4°C until use.

## Serology

The presence of anti-VP7 BTV antibodies in UV-light inactivated serum was investigated using the ID Screen Bluetongue Milk indirect kit (ID vet, Innovative Diagnostics; BTSMILK-4P) according to manufacturer's protocol. Sheep sera were diluted from 1:20 to 1:2560 and samples were tested in two technical replicates. Serum samples were tested on day 0 and 7 post infection. A BMG Labtech PHERAstar *FS* Elisa plate reader was used to obtain the data.

## Blood biochemistry and haematology

The blood was collected in vacuum tubes without anticoagulant for biochemical analysis, left at room temperature to allow clot retraction, and then centrifuged at 2000g for 4 minutes. Samples were analysed using the ILAB 650 automated system (Instrumentation Laboratory-Werfen, MA, USA) with the Quantilab Kits (Werfen Company, Milan, Italy) according to manufacturer's instructions. For the haematological analyses, tubes containing ethylenedi-aminetetraacetic acid (EDTA) at 4°C were analysed within 24 h after collection. Samples were

analysed with an ADVIA 120 haematology system (Siemens), equipped with specific software for veterinary use.

## Cytokines

To levels of cytokines in the sera of infected animals were obtained using the MILLIPLEX MAP Bovine Cytokine/Chemokine Magnetic Bead Panel 1—Immunology Multiplex Assay Kit (Merck-Millipore, BCYT1-33K) and the Bovine IFN-alpha ELISA Kit (Sigma-Aldrich, RAB1012) according to the manufacturer's instructions. Sera were diluted 1:2 and UV-inactivated before use.

## RNAseq

Two RNAseq analyses were carried out in this study on (i) RNA extracted from blood of infected and mock-infected animals (G1 and G2) at 1, 3 and 7dpi, and (ii) RNA extracted from immortalised ovine endothelial cells infected (or mock-infected) at 6, and 12 hpi. RNA from blood samples was isolated using the PAXgene Blood RNA Kit (Qiagen, 762164) according to manufacturer's instructions. Isolated RNA was stored at -80°C until processed. RNA concentration was assessed with a Qubit Fluorimeter (Lefe Technologies) while RNA integrity was calculated using an Agilent 4200 TapeStation. RNA from blood samples had an average RNA integrity value of 8 or above. From these samples, libraries for sequencing were generated using a Lexogen QuantSeq 3' mRNA-seq (FWD) kit, according to the manufacturer's instructions. Briefly, 35ng of total RNA from each sample was taken for library preparation. cDNA was synthesised directly from the polyadenylated 3' mRNA end with an oligodT. The RNA template was then enzymatically depleted and specific globin probes, that block the processing of alpha and beta haemoglobin sequences, were added to the cDNA. The primed cDNA was then converted to dsDNA by second strand synthesis using random primers containing Unique Molecular Identifiers (UMIs). Libraries were pooled in equimolar concentrations and sequenced in Illumina NextSeq 500 and 550 sequencers using a high-output cartridge, generating single reads with a length of 75 bp. At least 91% of the reads generated presented a quality score of 30 or above.

RNA of infected and mock-infected immortalised ovine endothelial cells was extracted as previously described [52] and had an average RNA Integrity Number of ~9.8. 500 ng of total RNA from each sample was taken for library preparation using an Illumina Stranded Total RNA Prep Ligation with Ribo-Zero Plus, according to the manufacturer's instructions. RNA was depleted of ribosomal RNA using specific probes and then fragmented. RNA fragments were reverse transcribed and converted to dsDNA, end repaired and A tailed. Samples were then ligated to adapters, followed by PCR amplification with indexing primers. Libraries were pooled in equimolar concentrations and sequenced in Illumina NextSeq 500 and 550 sequencers using a high-output cartridge, generating single reads with a length of 75 bp. At least 93% of the reads generated presented a quality score of 30 or above.

The quality of the RNAseq reads was assessed using FastQC (http://www.bioinformatics. babraham.ac.uk/projects/fastqc), and sequence adaptors were removed using TrimGalore (https://www.bioinformatics.babraham.ac.uk/projects/trim_galore). The reference *Ovis aries* genome (Oar_v3.1) wase downloaded from Ensembl and reads were subsequently aligned using Hisat2 [74] and counted using FeatureCount [75], respectively. The EdgeR package was then used to calculate the gene expression levels and to analyse differentially expressed genes [76]. For the *in vivo* experiments, in order to control for background inter-host variation we utilised the dream RNAseq differential expression method [77] implemented in R. This method allows individual-level animal variation to be controlled for using random effects in

order to better estimate the impact of treatment (in this case infection) as fixed effects. In order to provide as much data as possible for the model, to estimate the impact of infection we also utilised the uninfected control animals, again each with their own random effects for individual variation. We utilised separate runs to compare each dpi (1,3,7) to dpi 0 using the form 'form <- ~ Disease + (1|Individual)'.

The list of sheep interferon stimulated genes for downstream analysis was taken from Shaw et al. (2017) [52].

### BTV RT-PCR

To determine the amount of viral RNA in the blood of infected and mock-infected animals, RNA samples which were used for sequencing were used for the detection of segment 10 of BTV by RT-qPCR at indicated days post infection. The following probe: 5'-FAM-ARG-CTG-CAT-TCG-CAT-CGT-ACG- C-TAMRA-3', the forward primer: 5'-TGG-AYA-AAG--CRA-TGT-CAA-A-3' and the reverse primer 5'-ACR-TCA-TCA-CGA-AAC-GCT-TC-3' with a thermal profile of: 10 min 50°C, 2 min 95°C, 40x 10 sec 95°C and 30 sec 60°C with the Brilliant III Ultra-Fast qRT-PCR master mix (Agilent, 600884) were used according to manufacturer's protocol.

### Histology, immunohistochemistry and image analysis

2 to 4 um thick sections of formalin fixed and paraffin-embedded (FFPE) tissues from infected and mock-infected animals were cut and mounted on glass slides and stained with haematoxylin and eosin by standard procedures. For the detection of the BTV NS2 protein, a rabbit anti-NS2 antibody was used [78,79] in a dilution of 1:7000. Pre-treatment of tissue sections was carried out by pressure cooking in citrate buffer (pH 6.0) and the EnVision+ Single Reagent (HRP. Mouse, Agilent Technologies, K4001) was used as visualisation system in an autostainer (Autostainer Link 48, Agilent Technologies). As negative control, the primary antibody was replaced by mouse serum. Other antibodies used in the study were the following. Anti-Foxp3 FJK-16s antibody (eBioscience, Thermo Fisher Scientific, 14-5773-82; dilution 1:10); anti-CD3 (Agilent DAKO, A0452; dilution 1:100), anti Ki67 (Agilent DAKO, M724029-2, dilution 1:200), anti-pSTAT-1 (Cell Signalling, 9467S; dilution 1:200), anti-CD163 (Bio-Rad, MCA1853, dilution 1:100), anti-WC-1 (Bio-Rad, MCA838GA, dilution 1:20) and anti-Pax5 (Agilent DAKO, M7307; dilution 1:20). The VECTASTAIN Elite ABC HRP Kit (Peroxidase, Rat IgG; Vectorlabs, PK-6104) was used as a secondary system for detecting Foxp3, while for the other antibodies either the EnVision+/HRP, Mouse, HRP kit (Agilent DAKO, K400111-2) or the EnVision+/HRP, Rabbit, HRP kit (Agilent DAKO, K4003) as required. In all experiments, 3,3'-Diaminobenzidine (DAB) was used as a chromogen.

Immunostained slides were scanned with an Aperio VERSA 8 Brightfield, Fluorescence & FISH Digital Pathology Scanner (Leica Biosystems) at 200 x brightfield magnification. To detect the number of immune-positive cells, areas of the following tissues were manually outlined: tongue (excluding the epithelial mucosa), 3–4 pieces of lung tissue (1–2 pieces frontal lobes, 1–2 pieces distal lobes) per animal, infected skin (left forelimb), non-infected skin (dorsal back), lymph nodes of the infection site (left pre-scapular lymph nodes). Aperio Software ImageScope (Leica Biosystems) was used to automatically acquire the numbers of positive cells (NS2, CD3, Pax5, CD163, WC-1) or nuclei (Ki67, Foxp3) per total number of cells (NS2, CD3, Pax5, CD163, WC-) or nuclei (Ki67, Foxp3), respectively. In the skin, only the dermis was assessed, and data was analysed separately for superficial and deep dermis, while adipose tissue was excluded. For lymph node sections, only the cortex was investigated, while for the spleen, the connective tissue was excluded in the analysis. The whole liver section was instead

included. The algorithm and software settings were fine tuned for each organ. Subsequent analyses were carried out using the same setting for each organ. For those organs having more than two tissue sections, the mean value of the two sections was used in subsequent analyses. The mean values obtained from each organ of non-infected animals was used as background value (negative values were transformed to zero), and subtracted to the values obtained from each organ of the infected animals.

Absolute numbers of follicles per lymph node were derived from the right prescapular lymph nodes and were obtained from the average number of follicles obtained from 1 to 3 cross-sections per lymph node stained with HE.

### *In situ* hybridization

FFPE tissue sections were used for the detection of virus-specific RNA by *in situ*-hybridisation. The RNAscope 2.5 HD Reagent Kit-RED (code: 322350, Advanced Cell Diagnostics) and the probes V-BTV-1-ITL-2006KC3-seg4-C1 were designed and purchased (product code: 109711-C11; Advanced Cell Diagnostics). Positive and negative controls included a ubiquitin and a plant gene probe, respectively (codes: 310041 and 310043, Advanced Cell Diagnostics). Assays followed the manufacturer's instructions.

### Machine learning

The random forest method for machine learning was used to predict the virus and host parameters correlating with clinical outcome of BTV infection. Random forest was chosen because of the prioritisation metrics built in this method, which allow the most predictive parameters to be discovered and ranked. RandomForest and Caret R packages were used in order to build random forests for prediction (RandomForest function from RandomForest), and the recursive feature elimination algorithm was utilised to find the optimal number of parameters for analysis (rfe function from Caret). In order to find the most consistent predictive parameters, a custom cross-validation approach for predictions was implemented. Firstly, using the whole dataset, parameters were ranked using gini-importance and recursive feature elimination on the top 150 gini parameters were used to see how prediction accuracy varied with the number of parameters available to the random forest model. The value provided by the rfe algorithm was highly variable due to the inherent stochasticity in the algorithm. Hence, the value for parameter number N at the point that the initial slope of accuracy vs parameters began to taper off was chosen; this value was taught to capture the best trade off of accuracy using the fewest parameters. 500 runs of cross-validation were performed using two animals as testing data for each group. Within each of these cross-validation runs, the SMOTE method [80] was used to generate synthetic datapoints to balance predictive groups for the two imbalanced datasets (based on clinical score–"clinical state", or the groupings related to the virus used in the inoculum "four states of infection"). A random forest based on this data was then built, and the N top parameters in each run were recorded. Subsequently, the N parameters which appeared most often in the 500 runs were taken as a final list for that classification model. This list of N parameters was then used as the final set for 500 additional rounds of cross-validation training and testing with 2 holdout testing for each using the previously found N parameters. These runs also utilised SMOTE to balance for the two imbalanced datasets. This gave 1000 testing datapoints (2*500) for each group to show the accuracy of the machine learning predictions. The predictions made during these 500 rounds of cross validation are summarised in the confusion matrices, which show the animal's true state along the columns and predicted values along the rows (see Tables 1–3). To test that our predictions were not the result of over-fitting to the data, we also ran the machine learning modelling script with the rows shuffled, and the

individual animals' data randomised across samples and groups. The confusion matrices showing the lack of prediction power of these models can be found in S6, S7 and S8 Tables, while the ROC curves are shown in S3 Fig. In addition, we used the Boruta algorithm to confirm the data obtained by machine learning [41]. This algorithm creates a "shadow parameter" column for each parameter in the model, shuffling the values randomly across samples. The importance metrics are then calculated for the real and shadow parameters, and a parameter is deemed as "confirmed" if it shows higher importance values than these shuffled shadow parameters. Parameters are determined as "unimportant" if they produce values significantly lower than the maximum importance of the shadow parameters. If a parameter cannot be determined as confirmed or unimportant in the initial number of runs, it is marked as "tentative".

### Blood transcriptional modules

In order to combine our 6000 RNA-seq data points with our host and viral-related datapoints without disproportionately weighting the RNA seq results, we compressed the transcriptomics data into 247 blood transcriptional modules. We used the sheep mapped versions from Braun et al.[38] (generously provided by Artur Summerfield in.csv format). For each BTM, we took the geometric mean of each animal's different gene counts' proportion of total transcripts from that sample. These BTM representations of gene expression were further compressed to remove any fully redundant BTMs where one was a fully contained subset of genes of another BTM. We moved BTMs for which we had only one expressed gene in that BTM. We additionally recursively merged any BTMs which were more than 90% Pearson correlated across our dataset (script ~/scripts/R/sheep_mega_data/1.4.21/prep.BTMs.4.8.21.R). This produced a final set of 279* BTMs that we merged with our complete data set, providing a total of 332 parameters for analysis. In order to include postmortem parameters, we were forced to remove BTV-$1_{IT2013}$ animals from the machine learning analysis as they had recovered before these were collected.

### Heatmap plotting normalisation

We applied a normalisation per variable, where it's taking the log (x+1) of each value, minus the mean log(x+1) of just the control animals, then divided by the standard deviation of log(x +1) of all the (non-NA) values (in effect a log Z score, but only using the uninfected animals' mean values).

### Pathway analysis

Ensembl gene identifiers for each DEG of the blood transcriptome were converted to their respective human orthologue identifier and gene name. Up-regulated high confidence (FDR <0.05) DEGs identified from each pairwise comparison were used for differential pathway analysis in Metascape using *H. sapiens* for species analysis (https://metascape.org/gp/index. html#/main/step1) [81]. Pathway *p*-values <0.05 were considered significant for pathway enrichment. Heat maps were plotted in GraphPad Prism (version 10.2) as log2 mean counts per million (CPM). Missing gene values were plotted as zero.

### Statistical analysis

Graphs were created by using R version 4.3.2. or GraphPad Prism version 10.1.2. P-values<0.05 were considered as statistically significant.

## Supporting information

**S1 Fig. Clinical outcome of BTV-1$_{IT2006}$ infection in male and female sheep.** Three additional rams were experimentally infected in location G2 and compared to female sheep. (a) Clinical signs. Note significant difference in the severity of clinical signs in infected rams compared to ewes at 6 dpi (p = 0.0344; 2-way-ANOVA). (b) Rectal temperature. Only male animals infected with BTV-1 2006 have a fever peak at 3 dpi compared to all other groups; no significant differences between male and female animals infected with BTV-1 2006 have been detected at any time point. Data are shown as median, minimum and maximum values.
(TIF)

**S2 Fig. BTV infection induces the differential regulation of host pro-inflammatory immune defences in a strain specific manner.** RNA was extracted from BTV infected sheep at 7 days post infection (dpi) and used for RNA-Seq. Host reads were aligned to the sheep genome and normalised to their corresponding negative control group (counts per million; CPM). Differentially expressed genes (adjusted *p*-value < 0.05; FDR) were identified for each pairwise comparison (as indicated). (a) Metascape pathway analysis of up-regulated DEGs showing relative pathway enrichment for each pairwise comparison (*p*-value < 0.05; log10 *p*-values shown). Top 25 pathways shown (ranked on G1-BTV-1-2006). (b-d) Expression profile (mean log2 CPM) of host DEGs identified during BTV infection associated with DisGeNET inflammation (C0021368), vascular disease (C0042373), and immune suppression (C4048329) pathways. Missing values are plotted as zero. *P*-values and test shown.
(TIF)

**S3 Fig. Boruta method: Validation of data used for 332 parameters predictive for the disease states.** A) True and false positive rate is shown for six states of infection (a), for four states of infection (b) and clinical states (c).
(TIF)

**S1 Table. Animals and study design.**
(PDF)

**S2 Table. Values of the parameters used in this study (obtained at 7 days post-infection).**
(XLSX)

**S3 Table. Raw data of 50 parameters for 6 states of infection.**
(CSV)

**S4 Table. Raw data of 17 parameters for 4 states of infection.**
(CSV)

**S5 Table. Raw data of 100 parameters for clinical states of infection.**
(CSV)

**S6 Table. Boruta Method detected 60 parameters in total as important for defining the 4 states of infection, which confirmed 17 out of 17 detected with the Random forest.**
(XLSX)

**S7 Table. Boruta Method detected 93 parameters in total as important for defining the 6 states of infection, which confirmed 50 out of 50 detected with the Random forest.**
(XLSX)

**S8 Table. Boruta Method detected 64 parameters in total as important for defining the clinical states of infection, which confirmed 39 out of 100 detected with the Random**

**forest.**
(XLSX)

**S9 Table. Shuffled data on 6 stats of infection.**
(XLSX)

**S10 Table. Shuffled data on 4 stats of infection.**
(XLSX)

**S11 Table. Shuffled data on clinical stats of infection.**
(XLSX)

**S12 Table. Details of the 35 key parameters defining the clinical outcome of BTV infection.**
(XLSX)

**S13 Table. List of Log10 P-values for each virus-infected group for the pathway analysis used in S2A Fig.**
(XLSX)

**S14 Table. Pathway values and LogP and log (q-values) for the pathway analysis used in S2A Fig.**
(XLSX)

## Acknowledgments

The authors thank all team members of the Histology Research Service of University of Glasgow: Lynn Stevenson, Frazer Bell, Lynn Oxford, Jan Duncan, and Jessica Lee for the outstanding quality of their work in the histology lab. We also thank Giovanni Antonio Pilo for his invaluable support in the field work. The work would not have been possible without the invaluable help for animal care of Berardo De Dominicis, Doriano Ferrari, Massimiliano Caporale, Giampaolo Foschini in Teramo and the teams in Sassari, Italy.

## Author Contributions

**Conceptualization:** Vanessa Herder, Marco Caporale, Noemi Sevilla, Ciriaco Ligios, Massimo Palmarini.

**Data curation:** Vanessa Herder, Oscar A. MacLean, Xinyi Huang, Kyriaki Nomikou, Natasha Palmalux, Jenna Nichols, Rosario Scivoli, Chris Boutell, Jay Allan, Haris Malik, Georgios Ilia, Quan Gu, Gaetano Federico Ronchi, Daniela Antonucci, Sara Capista, Daniele Giansante, Ana Da Silva Filipe, Giantonella Puggioni, Meredith E. Stewart.

**Formal analysis:** Vanessa Herder, Marco Caporale, Oscar A. MacLean, Kyriaki Nomikou, Chris Boutell, Jay Allan, Haris Malik, Georgios Ilia, Quan Gu, Ana Da Silva Filipe, Giantonella Puggioni, Meredith E. Stewart.

**Funding acquisition:** Maria Teresa Mercante, Mauro Di Ventura, Ciriaco Ligios, Massimo Palmarini.

**Investigation:** Vanessa Herder, Marco Caporale, Oscar A. MacLean, Davide Pintus, Xinyi Huang, Kyriaki Nomikou, Natasha Palmalux, Jenna Nichols, Rosario Scivoli, Chris Boutell, Jay Allan, Haris Malik, Georgios Ilia, Quan Gu, Meredith E. Stewart.

**Methodology:** Vanessa Herder, Marco Caporale, Oscar A. MacLean, Davide Pintus, Xinyi Huang, Kyriaki Nomikou, Natasha Palmalux, Jenna Nichols, Chris Boutell, Quan Gu, Wilhelm Furnon, Sara Capista, Antonio Cocco, Meredith E. Stewart.

**Project administration:** Vanessa Herder, Marco Caporale, Davide Pintus, Aislynn Taggart, Gaetano Federico Ronchi, Ciriaco Ligios, Massimo Palmarini.

**Resources:** Marco Caporale, Wilhelm Furnon, Stephan Zientara, Emmanuel Bréard, Daniela Antonucci, Daniele Giansante, Maria Teresa Mercante, Mauro Di Ventura, Ciriaco Ligios.

**Software:** Oscar A. MacLean, Quan Gu.

**Supervision:** Vanessa Herder, Marco Caporale, Maria Teresa Mercante, Mauro Di Ventura, Ana Da Silva Filipe, Ciriaco Ligios, Massimo Palmarini.

**Validation:** Vanessa Herder, Oscar A. MacLean.

**Visualization:** Vanessa Herder, Oscar A. MacLean, Xinyi Huang, Chris Boutell.

**Writing – original draft:** Vanessa Herder, Oscar A. MacLean, Massimo Palmarini.

**Writing – review & editing:** Vanessa Herder, Marco Caporale, Oscar A. MacLean, Davide Pintus, Xinyi Huang, Kyriaki Nomikou, Natasha Palmalux, Jenna Nichols, Rosario Scivoli, Chris Boutell, Aislynn Taggart, Jay Allan, Haris Malik, Georgios Ilia, Quan Gu, Gaetano Federico Ronchi, Wilhelm Furnon, Stephan Zientara, Emmanuel Bréard, Daniela Antonucci, Sara Capista, Daniele Giansante, Antonio Cocco, Maria Teresa Mercante, Mauro Di Ventura, Ana Da Silva Filipe, Giantonella Puggioni, Noemi Sevilla, Meredith E. Stewart, Ciriaco Ligios, Massimo Palmarini.

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
