## [Decision Letter · Decision Letter 0]

18 May 2024

Dear Dr. Palmarini,

Thank you very much for submitting your manuscript "Correlates of Disease Severity in Acute Arbovirus Infection" for consideration at PLOS Pathogens. As with all papers reviewed by the journal, your manuscript was reviewed by members of the editorial board and by several independent reviewers. In light of the reviews (below this email), we would like to invite the resubmission of a significantly-revised version that takes into account the reviewers' comments.

Please address the robustness of the model used in feature selection.

We cannot make any decision about publication until we have seen the revised manuscript and your response to the reviewers' comments. Your revised manuscript is also likely to be sent to reviewers for further evaluation.

Sincerely,

Xiu-Feng Wan

Guest Editor

PLOS Pathogens

Guangxiang Luo

Section Editor

PLOS Pathogens

Michael Malim

Editor-in-Chief

PLOS Pathogens

orcid.org/0000-0002-7699-2064

Please address the robustness of the model used in feature selection.

Reviewer's Responses to Questions

**Part I - Summary**

Reviewer #1: Herder et al. propose a data-driven study to identify the parameters related to acute arbovirus infection. This manuscript uses the recurrent feature selection approaches with Gini Importance in the random forest model to identify a set of parameters prone to classify known phenotypical information. These parameters are inferred to be related to the severity of arboviral disease. Using data-driven approaches is highly appreciated in the field, as they can identify some factors purely unbiasedly. However, from the perspective of machine learning and data analysis, some points need to be clarified or strengthened.

Major:

1. Line 178 ‘We used recursive feature elimination to find the most predictive 179 core subset of parameters distinguishing each group’. A quantitative measurement, i.e., slope or something, or a kind of grid search, needs to be adopted to increase the scientific rigor of the studies. From the naked eye, from Figure 2, it is arbitrary to set up these thresholds to select the top parameters of the model. The same issue with many ‘arbitrary’ numbers in the study, such as Line 182, ‘…required a 182 minimum of 50…’, why 50?

2. Line 187, ‘We used 1000 rounds of cross-validation, each using randomly selected 5 animals from each class to train the model and 2 unseen animals’. Cross validation is usually used to check the model’s performance, the proportion of these 1000 does not necessarily reveal the feature and data’s performance and importance. Instead, a confusion matrix, ROC curve, and precision-recall curve can be used to quantify the performance and robustness.

3. It may be good to have some other algorithms other than Random Forest to check the robustness of the feature selected. For example, Boruta (https://cran.r-project.org/web/packages/Boruta/Boruta.pdf) may be a good candidate.

Minor:

1. Line 264: Importantly, analysis of blood transcriptome using standard pathway analysis methods also revealed “cytokine signaling in immune system” (GO:0019221), “innate immune responses” (GO:0002226) and “interferon signalling” (GO:0060337). It could be better to have pvalue or FDR here to demonstrate the significance of GO terms

2. Typo in Short Title: ‘…in Arboviral dDsease’

Reviewer #2: Herder and colleagues described correlates of disease severity in acute bluetongue virus (BTV) infection in sheep. Authors used three bluetongue viruses showing different clinical signs to infect sheep and then employed machine learning to identified determinants of virus and host associated with disease pathogenesis. They identified 5 factors critical for clinical disease and pathogenesis. Then authors conclude that using an agnostic machine learning approach is a good tool to understand pathogenetic mechanisms affecting the disease outcome of an arbovirus infection. This manuscript is well-written and this study providing some interesting results.

**Part II – Major Issues: Key Experiments Required for Acceptance**

Reviewer #1: 1. Line 178 ‘We used recursive feature elimination to find the most predictive 179 core subset of parameters distinguishing each group’. A quantitative measurement, i.e., slope or something, or a kind of grid search, needs to be adopted to increase the scientific rigor of the studies. From the naked eye, from Figure 2, it is arbitrary to set up these thresholds to select the top parameters of the model. The same issue with many ‘arbitrary’ numbers in the study, such as Line 182, ‘…required a 182 minimum of 50…’, why 50?

2. Line 187, ‘We used 1000 rounds of cross-validation, each using randomly selected 5 animals from each class to train the model and 2 unseen animals’. Cross validation is usually used to check the model’s performance, the proportion of these 1000 does not necessarily reveal the feature and data’s performance and importance. Instead, a confusion matrix, ROC curve, and precision-recall curve can be used to quantify the performance and robustness.

3. It may be good to have some other algorithms other than Random Forest to check the robustness of the feature selected. For example, Boruta (https://cran.r-project.org/web/packages/Boruta/Boruta.pdf) may be a good candidate.

Reviewer #2: 1) The title looks like not fit very well as only BTV was tested in one species , not other arboviruses. Not sure whether the same results will be obtained in other viruses in different species.

2) Based on the results, the sex is critical for clinical disease, especially the male sheep show more severe disease than the female. Although author tested 332 biological parameters, likely no clear explanations on it and suggest discussing it.

3) The study design did not consider the animal age. Normally BTVs can cause more severe disease in young animals. Should discuss it what will be turned out if young animals used.

**Part III – Minor Issues: Editorial and Data Presentation Modifications**

Reviewer #1: 1. Line 264: Importantly, analysis of blood transcriptome using standard pathway analysis methods also revealed “cytokine signaling in immune system” (GO:0019221), “innate immune responses” (GO:0002226) and “interferon signalling” (GO:0060337). It could be better to have pvalue or FDR here to demonstrate the significance of GO terms

2. Typo in Short Title: ‘…in Arboviral dDsease’

Reviewer #2: (No Response)

PLOS authors have the option to publish the peer review history of their article (what does this mean?). If published, this will include your full peer review and any attached files.

Reviewer #1: No

Reviewer #2: No
---

## [Decision Letter · Decision Letter 1]

31 Jul 2024

Dear Prof. Palmarini,

We are pleased to inform you that your manuscript 'Correlates of Disease Severity in Bluetongue as a Model of Acute Arbovirus Infection' has been provisionally accepted for publication in PLOS Pathogens.

Best regards,

Xiu-Feng Wan

Guest Editor

PLOS Pathogens

Michael Letko

Section Editor

PLOS Pathogens

Michael Malim

Editor-in-Chief

PLOS Pathogens

orcid.org/0000-0002-7699-2064

Reviewer Comments (if any, and for reference):

Reviewer's Responses to Questions

**Part I - Summary**

Reviewer #1: The authors significantly improved the manuscript. No more questions.

**Part II – Major Issues: Key Experiments Required for Acceptance**

Reviewer #1: NA

**Part III – Minor Issues: Editorial and Data Presentation Modifications**

Reviewer #1: NA

PLOS authors have the option to publish the peer review history of their article (what does this mean?). If published, this will include your full peer review and any attached files.

Reviewer #1: No

---

## [Editor Report · Acceptance letter]

10 Aug 2024

Dear Prof. Palmarini,

We are delighted to inform you that your manuscript, "Correlates of Disease Severity in Bluetongue as a Model of Acute Arbovirus Infection," has been formally accepted for publication in PLOS Pathogens.

Best regards,

Michael Malim

Editor-in-Chief

PLOS Pathogens

orcid.org/0000-0002-7699-2064